



# Elemental analysis of Oxygenated Organic Coating on Black Carbon Particles using a Soot-Particle Aerosol Mass Spectrometer

Mutian Ma[1], Laura-Hélèna Rivellini[2], YuXi Cui[1], Megan D. Willis[3], Rio Wilkie[4], Jonathan P. D. Abbatt[4], Manjula R. Canagaratna[5], Junfeng Wang[6], Xinlei Ge[6], Alex K.Y. Lee[1,2]

[1] Department of Civil and Environmental Engineering, National University of Singapore, Singapore
[2] NUS Environmental Research Institute, National University of Singapore, Singapore
[3] Lawrence Berkeley National Lab, Chemical Sciences Division, Berkeley, CA, USA
[4] Department of Chemistry, University of Toronto, Toronto, ON, Canada
[5] Aerodyne Research, Inc., Billerica, MA, USA
[6] School of Environmental Science and Engineering, Nanjing University of Information Science and Technology, Nanjing, China

*Correspondence to*: Alex K. Y. Lee (ceelkya@nus.edu.sg)

**Abstract**

Chemical characterization of organic coatings is important to advance our understanding of the physio-chemical
properties and environmental fate of black carbon (BC) particles. The soot-particle aerosol mass spectrometer (SP-AMS) has been utilized for this purpose in recent field studies. The laser vaporization (LV) scheme of SP-AMS can heat BC cores gradually until they are completely vaporized, during which organic coatings can be vaporized at temperatures lower than that of the thermal vaporizer (TV) used in a standard high-resolution time-of-flight aerosol mass spectrometer (HR-ToF-AMS) that employs flash vaporization. This work investigates the effects of vaporization
schemes on fragmentation and elemental analysis of known oxygenated organic species using three SP-AMS instruments. We show that LV can reduce fragmentation of organic molecules. Substantial enhancement of $C_2H_3O^+/CO_2^+$ and $C_2H_4O_2^+$ signals was observed for most of the tested species when the LV scheme was used, suggesting that the observational frameworks based using HR-ToF-AMS field data may not be directly applicable for evaluating the chemical evolution of oxygenated organic aerosol (OOA) components coated on ambient BC particles.
The uncertainties of H:C and O:C determined by the improved-ambient (I-A) method for both LV and TV approaches were similar, with scaling factors of 1.10 for H:C and 0.89 for O:C were determined to facilitate more direct comparisons between observations from the two vaporization schemes. Furthermore, the I-A method was updated based on the multilinear regression model for the LV scheme measurements. The updated parameters can reduce the relative errors of O:C from -26.3% to 5.8%, whereas the relative errors of H:C remain roughly the same. Applying the
scaling factors and the updated parameters for the I-A method to ambient data, we found that even though the time series of OOA components determined by the LV and TV schemes are strongly correlated at the same location, OOA coatings were likely less oxygenated compared to those externally mixed with BC.



## 1 Introduction

Atmospheric black carbon (BC) particles have significant impacts on climate and human health. Studies have shown that BC particles have negative impact on vascular, cardiopulmonary, respiratory and chronic diseases (Brook et al., 2010; Heal et al., 2012; Heinzerling et al., 2016; Nel, 2005). They are strongly light-absorbing, resulting in positive

radiative forcing which is equivalent to ~55% of the radiative forcing caused by carbon dioxide (Bond and Bergstrom, 2007; Bond et al., 2013; Ramanathan and Carmichael, 2008). Ambient BC particles are largely internally mixed with organic aerosols (OA). While BC and hydrophobic organic coating can be co-emitted from combustion processes, BC can be coated by oxygenated organic compounds formed via oxidation of organic vapors and heterogeneous processing (China et al., 2013; Lee et al., 2019; Wu et al., 2019). Organic coatings can alter the physio-chemical

properties of BC particles, such as enhancements of light absorptivity (Liu et al., 2017; Peng et al., 2016), cloud formation potential (Kuwata et al., 2009; Liu et al., 2013) and subsequently their atmospheric transport and lifetime (Bond et al., 2013; Laborde et al., 2013; McMeeking et al., 2011). However, characterization of organic coatings on BC particles remains a great analytical challenge due to their chemical complexity.

The soot-particle aerosol mass spectrometer (SP-AMS), which is a standard high-resolution time-of-flight aerosol mass spectrometer (HR-ToF-AMS) equipped with an additional infrared (IR) laser vaporizer (Onasch et al., 2012), has been utilized to characterize refractory BC (rBC) and its coatings (Collier et al., 2018; Lee et al., 2017; Massoli et al., 2015; Willis et al., 2016; Wu et al., 2019). The laser vaporization scheme of SP-AMS can heat up rBC cores gradually (up to ~4000K) until they are completely vaporized, during which non-refractory coatings can be vaporized

at a temperature lower than the operating temperature of thermal vaporizer (600°C) typically used in HR-ToF-AMS for flash vaporization (DeCarlo et al., 2006; Onasch et al., 2012) (Figure 1). Different fragmentation patterns of OA have been observed between the two vaporization schemes in both laboratory and field studies (Canagaratna et al., 2015b; Massoli et al., 2015). In particular, positive matrix factorization (PMF) of field measurements has shown that mass spectra of oxygenated OA (OOA), which usually represents secondary OA (SOA) components in the

atmosphere, determined by the laser vaporization (LV) scheme of SP-AMS were dominated by $C_2H_3O^+$ signals (Collier et al., 2018; Wu et al., 2019). In contrast, the co-located HR-ToF-AMS measurements (i.e., thermal vaporization (TV) scheme) generated mass spectra of more-oxidized OOA components with the most intense signals from $CO^+$ and $CO_2^+$ fragments (Chen et al., 2018; Lee et al., 2017; Massoli et al., 2015) as illustrated in Figure 1a and 1b.


Elemental ratios of OA (i.e., hydrogen-to-carbon, H:C and oxygen-to-carbon, O:C) measured by HR-ToF-AMS have been widely used to investigate the physical and chemical properties of OA (Koop et al., 2011; Kuwata et al., 2012; Lambe et al., 2013; Massoli et al., 2010; Wong et al., 2011) and their evolution depends on the types of primary emissions and aging processes (Heald et al., 2010; Jimenez et al., 2009; Kroll et al., 2011). The elemental ratios

obtained from HR-ToF-AMS mass spectra can be potentially biased by vaporization and ion fragmentation processes as described in detail by Canagaratna et al. (2015a). To account for such measurement uncertainties, the calibration





factors between experimentally measured and theoretical elemental compositions of known organic compounds were reported by Aiken et al. (2007; 2008). The "Aiken-Ambient" (A-A) method was developed for elemental analysis of ambient OA using empirically estimated $CO^+$ and $H_2O^+$ ion signals to avoid interferences from ambient air (Aiken et al., 2008). Canagaratna et al. (2015a) further established the "Improved-Ambient" (I-A) method that uses specific ion

fragments as markers to reduce composition-dependent systematic biases. Both A-A and I-A methods have been fully integrated to the standard procedure for analyzing ambient OA measured by HR-ToF-AMS. However, the two methods were developed based on OA mass spectra generated by the TV approach, hence their direct applications for determining elemental compositions of OA vaporized by the LV scheme (i.e., organic coating on BC particles) may not be appropriate (Canagaratna et al., 2015b).

To improve the accuracy of elemental analysis for OOA materials coated on ambient BC particles using SP-AMS, this work compares the elemental ratios (H:C and O:C) of known oxygenated organic compounds determined by both TV and LV schemes. The I-A method was applied to determine the elemental composition of OOA materials for all our laboratory experiments. Such comparison has been conducted by Canagaratna et al. (2015b), in which only a small

number of organic species was tested by a single SP-AMS. In this study, we extend this type of investigation in three ways: (1) increasing the number of organic species with different functional groups to be tested, (2) deploying two additional independent SP-AMS from different research groups to conduct our measurements, and (3) generating new fitting parameters based on the approach for developing the I-A method to enhance the accuracy of elemental analysis for organic coatings detected by the LV scheme of SP-AMS. The results can be used to evaluate the robustness of

applying the combination of the laser vaporization approach and the I-A method for determining H:C, O:C and $OS_C$ of ambient OOA coated on BC particles, and to provide insight into the potential discrepancies between ambient OOA materials that are externally and internally mixed with BC particles.

## 2.    Experiment

### 2.1  Particle generation

A total of 30 oxygenated organic species, including dicarboxylic acids, polycarboxylic acids, alcohols, and multifunctional compounds, was included in this study (Table 1). Small amounts of a standard organic compound were dissolved in ultrapure water, which was subsequently used to generate pure organic particles using a constant output atomizer (Model 2076, TSI). For generating rBC-organic mixed particles, Regal Black (Regal 400R pigment,

Cabot Corp) and a standard organic compound were mixed in the bulk solution for atomization. Regal Black was used in this study because it has been suggested as an rBC standard for calibrating the LV scheme of SP-AMS (Onasch et al., 2012). Atomized particles were subsequently dried by a diffusion dryer using silica gel to minimize the interference of particle-phase water to $H_2O^+$ signals and other related fragments (i.e., $O^+$ and $HO^+$). Pure argon gas was used for


atomization and dilution to minimize the interference of gaseous $N_2$ and $CO_2$ to the quantification of $CO^+$ and $CO_2^+$ signals in organic mass spectra (Canagaratna et al., 2015a; Corbin et al., 2014; Willis et al., 2014).

## 2.2  Soot-particle aerosol mass spectrometer measurements

The details of SP-AMS (Aerodyne Research, Inc.) have been reported in detail by Onasch et al. (2012). In brief, the SP-AMS is equipped with a thermal vaporizer (i.e., a heated tungsten surface) and a laser vaporizer (i.e., a continuous wave intra-cavity 1064 nm Nd:YAG laser). While the thermal vaporizer operated at 600°C can vaporize non-refractory particulate matter (NR-PM, including organic, sulfate, nitrate, ammonium and chloride), the laser vaporizer is designed for vaporizing rBC-containing particles at which rBC cores can be gradually heated up to ~4000K. During

this heating process, organic coatings can be vaporized at a lower range of temperature (likely < 600°C) depending on the volatility of each organic compound. The vaporized analytes are ionized using 70 eV electron impact (EI) ionization and the ions are subsequently detected by a high-resolution time-of-flight mass spectrometer operated in V-mode (Canagaratna et al., 2007; DeCarlo et al., 2006).

The SP-AMS instruments were operated in two different vaporization schemes for characterizing pure organic particles and rBC-organic mixed particles, respectively. When the laser vaporizer of the SP-AMS was off, the instrument was operated as a standard HR-ToF-AMS to facilitate flash-vaporization of pure organic particles (TV scheme). For the second part of the experiments, the thermal vaporizer was removed from SP-AMS and the laser vaporizer was turned on for measuring standard organic compounds coated on Regal Black exclusively (LV scheme,

Figure 1c). Note that some pure organic particles might be generated through atomization of the rBC-organic mixture but they cannot be detected by the LV scheme. Observations from three SP-AMS were reported in this study and they are labeled as SP-AMS 1, 2, and 3. SP-AMS 1 and 2 were used to generate new data for 18 and 20 organic species, respectively. Data of SP-AMS 3 (10 organic species) were extracted from Canagaratna et al. (2015b). Table 1 summarizes the tested species for each SP-AMS. The three SP-AMS were operated by different researchers from the

National University of Singapore (SP-AMS 1), University of Toronto (SP-AMS 2) and Aerodyne Research (SP-AMS 3).

## 2.3  Data analysis

The raw data of SP-AMS measurements were processed by the AMS data analysis software (Squirrel for unit mass resolution (UMR) data and PIKA 1.21b for high-resolution peak fitting, (Sueper, 2019)) and statistical analysis were

processed by R (version 3.6). Given that pure argon gas was used for particle generation and dilution in all the experiments, $CO^+$ signals could be quantified in the high-resolution aerosol mass spectra. Note that only small $N_2^+$ signals were detected in some of the experiments using SP-AMS 1 due to the residual $N_2$ desorbed from the desiccant. For the results of rBC-organic mixed particles, the interference of refractory $CO_x^+$ signals (r$CO_x$, i.e., $CO^+$ and $CO_2^+$)





formed during rBC vaporization to organic mass spectra were corrected in the fragmentation table using $C_1^+:C_3^+$, $CO^+:C_3^+$ and $CO_2^+:C_3^+$ ratios obtained from pure Regal Black particles. Elemental analysis was performed using the I-A method (Canagaratna et al., 2015b) to calculate H:C and O:C ratios of each organic compound. Average carbon oxidation state of each organic compound was also calculated based on the method reported in Kroll et al. (2011) (i.e.,

$OS_c = 2 \times O:C - H:C$).

## 3.  Results and discussion

### 3.1  Different fragmentation patterns between TV and LV schemes

Figure 2 shows the normalized mass spectra of azelaic acid (a dicarboxylic acid) measured by the SP-AMS 1 for illustrating different fragmentation patterns generated by the TV and LV schemes. The comparison for azelaic acid shows that the LV approach can produce larger organic fragments (e.g., m/z 60, 69, 73, 83 and 84) compared to the TV approach. Similarly, arabitol (a sugar alcohol) and levoglucosan (a chemical marker for biomass burning OA)

show less fragmentation when they were vaporized with rBC particles by the laser vaporizer (Figures S1 and S2). The normalized cumulative histograms of m/z for the organic compounds measured by the SP-AMS 1 are presented in Figure 2c (i.e., the average of 18 species), which clearly shows that the curve shifts toward larger m/z when the compounds were vaporized by the LV scheme. The green dashed line represents the differences between respective fragments from the TV (blue line) and LV (red line) schemes. It shows a decreasing trend within the red shaded region,

illustrating that the LV scheme could generate more organic fragments starting from m/z 55 on average. Overall, our observations support the general hypothesis and previous observations (Canagaratna et al., 2015b) that a thermal vaporizer operated at 600°C tends to generate organic mass spectra with smaller molecular fragments compared to the LV scheme, in which organic vaporization and fragmentation can occur at a lower range of temperature, for the same organic compound.

### 3.2  Changes in $f_{C_2H_3O^+}/f_{CO_2^+}$ ratio

The organic fragments of $C_2H_3O^+$ and $CO_2^+$ are the two dominant peaks observed in ambient OOA components identified by the PMF analysis of standard HR-ToF-AMS measurements (i.e., TV scheme) (Ng et al., 2011). The

relative importance of the two fragments varies between OOA components identified at the same locations, and the ratio of $f_{C_2H_3O^+}/f_{CO_2^+}$ ($f_{i^+}$= a mass fraction of m/z $i^+$ to total organic) usually decreases with the degree of oxidative aging (Ng et al., 2011; Ng et al., 2010). Based on the observations from SP-AMS 1 and 2, Figure 3a shows that most of the organic species detected by the LV scheme gave higher $f_{C_2H_3O^+}/f_{CO_2^+}$ ratios compared to the TV scheme. Note that dicarboxylic acids and multifunctional organic compounds show stronger enhancement compared to alcohols.



The average of $f_{C_2H_3O^+}/f_{CO_2^+}$ ratios measured by the TV and LV schemes are 0.48 (± 0.52) and 3.07 (± 3.59), respectively, indicating less thermal-induced decarboxylation with the gradual vaporization.

The observed enhancement of $f_{C_2H_3O^+}/f_{CO_2^+}$ ratios in this work suggest that the f44 vs. f43 (or $f_{CO_2^+}$ vs. $f_{C_2H_3O^+}$) observational framework developed based on the HR-ToF-AMS datasets worldwide by Ng et al. (2010) may not be directly applicable for evaluating the degree of aging and characteristics of OOA coatings measured by the LV scheme of SP-AMS. Figure 3a compares the ambient OOA components determined by the TV and LV schemes (i.e., concurrent HR-ToF-AMS and SP-AMS LV scheme measurements) at the same location if their time series of mass concentrations are strongly correlated (i.e., R > 0.75). Table S1 summaries the characteristics of the OOA components observed in Beijing summer (Xie et al., 2019a; Xu et al., 2019), Beijing winter (Wang et al., 2019; Xie et al., 2019b), Tibet (Wang et al., 2017; Xu et al., 2018), and Fontana, CA (Chen et al., 2018; Lee et al., 2017) that are used for our comparison. It can be found that most of the OOA factors determined by the LV scheme gave significantly higher $f_{C_2H_3O^+}/f_{CO_2^+}$ ratios compared to those determined by the TV scheme by factors of 6-18, which is similar to those observed for dicarboxylic acids and multifunctional organic compounds (Figure 3a). Therefore, the differences in $f_{C_2H_3O^+}/f_{CO_2^+}$ ratios between the ambient OOA components determined by the two vaporization can be partially explained by the laboratory observation reported in this Section. Nevertheless, we cannot rule out the possibility that the chemical compositions of the OOA coatings on BC particles were different than those externally mixed with BC particles despite their strong temporal correlations (see more discussion on this topic in Section 3.6 based on the elemental analysis).

### 3.3 Changes in $f_{C_2H_4O_2^+}$

$C_2H_4O_2^+$ is a tracer fragment often associated with biomass burning OA (BBOA) and $f_{C_2H_4O_2^+}$ is commonly used to identify the presence of BBOA and to evaluate its degree of aging (Bozzetti et al., 2017; Cubison et al., 2011; Milic et al., 2017). Figure 3b shows that most of the species from SP-AMS 1 and 2 (i.e., 97% of the tested species) showed the enhancement of $f_{C_2H_4O_2^+}$ when they were detected by the LV scheme regardless of their functional moieties. The average enhancement factor of $f_{C_2H_4O_2^+}$ for alcohol, dicarboxylic acids, and multifunctional groups are 2.62±0.92, 2.90+0.78, and 2.69±1.11, respectively. Levoglucosan, a known cellulose-derived compound produced during biomass burning (Simoneit et al., 1999), gives an enhancement of $f_{C_2H_4O_2^+}$ by a factor 2.33. Different to $f_{C_2H_3O^+}/f_{CO_2^+}$ ratios that show scattered data between vaporization schemes, a strong linear correlation was obtained for all the tested species (R = 0.95) with the slope equal to 2.45. This strong linear correlation suggests that the $f_{C_2H_4O_2^+}$ increased in similar extent for most of the tested oxygenated organic species.



Together with the changes in $f_{CO_2^+}$ caused by the vaporization schemes, our observation suggests that the f44 vs. f60 (or $f_{CO_2^+}$ vs. $f_{C_2H_4O_2^+}$) observational framework developed by Cubison et al. (2011) has to be used cautiously for evaluating potential influences of biomass burning emissions on the chemical composition of OOA coatings. For example, Rivellini et al. (2019) observed that the laser vaporization approach lead to the enhancement of $f_{C_2H_4O_2^+}$ for

total OA, LO-OOA and MO-OOA in an urban environment by comparing their laser-off and laser-on measurements using a dual-vaporizers scheme (i.e., the SP-AMS switched between the TV scheme and TV + LV scheme during operation). Wang et al. (2019) and Xie et al. (2019b) identified BBOA factors in Beijing winter from their concurrent SP-AMS and HR-ToF-AMS measurements, respectively, and higher $f_{C_2H_4O_2^+}$ value was observed for the BBOA factor determined by the TV scheme. However, it is important to note that the two BBOA factors were weakly

correlated (R = 0.42), and thus they likely represented BBOA materials from different origins.

### 3.4 Elemental analysis of organic coating

Elemental analysis of pure OA and organic coatings on rBC particles was performed based on the I-A method. Tables S2-S4 summarize the H:C, O:C and $OS_C$ values of all the organic compounds measured by the three SP-AMS. Figure

4 compares the measured H:C, O:C and $OS_C$ values of all the organic compounds generated by the LV scheme to their true values. While the measured H:C ratios scattered around the 1:1 line (Figure 4a), the measured O:C and $OS_C$ values are generally lower than their corresponding true values (Figure 4b and 4c). The average relative errors of H:C and O:C ratios for individual SP-AMS varied from –3.4% to 11.5% (mean = 6.6%) and from –37.1% to –22.0% (mean = –26.3%), respectively (Table 2). Note that there are no statistical differences (ANOVA, p < 0.05) between the relative

errors of elemental ratios determined by the three instruments, suggesting that the elemental analysis is not strongly instrument dependent. For the thermal vaporization approach, the average relative errors of H:C (mean = –5.2%) and O:C (mean = –21.5%) ratios determined in this work are similar to the measurement uncertainties of HR-ToF-AMS previously reported by Canagaratna et al. (2015a).

Figure 5 compares the elemental ratios of standard organic compounds determined by the LV and TV schemes. The data points from the three independent SP-AMS are well-aligned with each other even though a large fraction of the tested organic species (~50%) were not repeated between the different SP-AMS instruments. The linear fits of all the measured data demonstrate that O:C and H:C ratios determined by the LV scheme differ from their corresponding values determined by the TV scheme by factors of 0.89 and 1.10, respectively. Canagaratna et al. (2015b) conducted

similar comparisons for 10 organic species (i.e. data from SP-AMS 3 in this work) based on the results obtained from the A-A method, reporting that the O:C and H:C values determined by the LV scheme differ from their corresponding values determined by the TV scheme by factors of 0.83 and 1.16, respectively. The uncertainty of O:C and H:C ratios are further reduced to approximately ±10% between the TV and LV schemes on average in this study. Figure S5 shows that the I-A method with the scaling factor applied can improve the accuracy of elemental ratios of oxygenated

species in general except for the H:C ratios of alcohol group. While the fragmentation of oxygenated organic species



due to the TV and LV scheme can be significantly different, this work illustrates that their elemental compositions can be comparable to the I-A method applied for laboratory-generated particles with a single oxygenated organic species.

**3.5  Improved-Ambient method for the LV scheme (I-A$_{sp}$)**

The I-A method has been widely used for the elemental analysis of ambient OA measured by the TV scheme of HR-ToF-AMS and this approach has been described in detail by Canagaratna et al. (2015a). In brief, chemical standards, including dicarboxylic acids, multifunctional acids and alcohols, were tested using the TV scheme. The elemental ratios were first determined by the A-A method developed by Aiken et al. (2008) and they were subsequently corrected

by a multi-linear regression (MLR) model based on the fraction contributions of $CHO^+$ and $CO_2^+$ fragments (i.e., $f_{CHO^+}$ and $f_{CO_2^+}$) to address the composition-dependence in the OA fragmentation, which is referred to as I-A method. Note that $f_{CHO^+}$ and $f_{CO_2^+}$ can be used as surrogates for alcohol and acid groups, respectively, and they are major peaks observed in ambient OOA (Canagaratna et al., 2015a; Duplissy et al., 2011; Takegawa et al., 2007). As illustrated in the previous Sections, OA fragmentation can be significantly different between the LV and TV schemes, and hence

an updated multilinear regression parameter is conducted to check whether the model accuracy can be improved for analyzing data generated by the LV scheme. Following the approach for developing the I-A method (Canagaratna et al., 2015a), updated multiple linear regression parameters for determining H:C and O:C ratios of organic coatings were obtained based on the data from two SP-AMS (1 and 2) as shown in Equation 1 and 2, respectively.

$$H{:}C_{I-A_{SP}} = H{:}C_{A-A} \times [0.90 + 1.02 \times f_{CHO^+} + 2.78 \times f_{CO_2^+}] \qquad \text{(Eq.1)}$$
$$O{:}C_{I-A_{SP}} = O{:}C_{A-A} \times [1.74 - 2.50 \times f_{CHO^+} + 1.93 \times f_{CO_2^+}] \qquad \text{(Eq. 2)}$$

where $H{:}C_{I-A_{SP}}$ and $O{:}C_{I-A_{SP}}$ are the elemental ratios obtained from the new fitting parameters (denoted as I-A$_{sp}$ method hereafter) and $H{:}C_{A-A}$ and $O{:}C_{A-A}$ are the elemental ratios obtained by the A-A method.

Given that most of the tested organic species were not the same for each SP-AMS, direct inter-instrument comparison was not possible. Hence the average relative errors obtained from the two instruments were used to evaluate the performance of I-A$_{SP}$ method across all the tested species. The relative errors of I-A$_{sp}$ method for H:C, O:C and OS$_C$ were 6.3%, 5.8%, and -9.8%, respectively, for the LV scheme data. While the I-A$_{sp}$ method leads to substantial

improvement of the relative errors of O:C compared to the I-A method (i.e., from -26.3% to 5.8%), the relative average errors of H:C ratio obtained from I-A$_{sp}$ and I-A method are comparable. As shown in Figure S3, the H:C, O:C and OS$_C$ values calculated by the I-A$_{sp}$ method are better aligned with the 1:1 line compared to those determined by the I-A method (Figure 4). However, it is worth noting that the I-A$_{sp}$ method gives the highest positive bias for the O:C ratio for alcohol species compared to the results from the I-A method (with and without applying the scaling factor)



as illustrated in Figure S5. For the H:C ratios, the I-A$_{sp}$ method can reduce relative error for alcohol species but generate larger range of errors for dicarboxylic acids and multifunctional species.

## 3.6 Insight into ambient OOA characteristics

There is an increasing number of field studies operating a standard HR-ToF-AMS and an SP-AMS concurrently (i.e., total OA measured by the TV scheme vs. organic coatings measured by the LV scheme) to investigate the mixing state of BC particles and the effects of primary emissions and atmospheric processing on the formation of organic coatings on BC particles (Lee et al., 2017; Massoli et al., 2015; Wang et al., 2020). In particular, whether SOA materials condensed on BC particles have similar chemical characteristics to those externally mixed with BC particles remains poorly understood. Previous observations in urban environments have reported that the mass spectral features of ambient OOA components identified by the PMF analysis were significantly different between the two co-located measurements even though their temporal variabilities strongly correlated to each other (Chen et al., 2018; Lee et al., 2017; Liu et al., 2019; Massoli et al., 2015; Wang et al., 2020; Xu et al., 2019; Zhao et al., 2019). Given our observations that vaporization scheme plays a critical role in the fragmentation process of oxygenated organic species, the LV elemental analysis scaling factors (0.89 for H:C and 1.10 for O:C) and the I-A$_{sp}$ method obtained in this work can facilitate more direct and robust comparison between the two types of measurements based on elemental analysis.

Figure 6 shows the elemental ratios of OOA components observed from previous field studies conducted in California Research at the Nexus of Air Quality and Climate Change (CalNex) 2010 campaign (Massoli et al., 2015), Fontana, California in 2015 (Chen et al., 2018; Lee et al., 2017), Tibet in 2015 (Wang et al., 2017; Xu et al., 2018), Beijing winter in 2016 (Wang et al., 2019; Xie et al., 2019b), and Beijing summer in 2017 (Xie et al., 2019a; Xu et al., 2019). Correlations of hourly-averaged mass concentrations between different PMF factors identified by the TV and LV schemes were investigated and only strongly correlated OOA factors (R > 0.75) were included in Figure 6. If an OOA component correlated well to multiple PMF OA factors identified by another vaporization scheme, the comparison was only performed for a pair of OA factors that gave the strongest correlation (see Table S1). Note that a transported BBOA factor identified in Tibet, which was associated with a significant amount of OOA materials (R = 0.96), was also included in this comparison. Figure 6a and 6b shows the adjusted H:C and O:C ratios measured by the LV scheme (y-axis) using the inter-conversion factor and the I-A$_{sp}$ method, respectively. The error bars of each data point represent the average absolute errors of the I-A and I-A$_{sp}$ method obtained in this study. Note that both I-A and I-A$_{sp}$ methods likely over-estimate O:C ratios of organics coated on ambient BC particles because of the contributions of refractory CO$_x^+$ fragments (e.g. CO$^+$ and CO$_2^+$) usually remain unresolved.

In Figure 6a, although a few H:C and O:C ratios were well-aligned onto the 1:1 line, it can be found that the majority of the O:C ratios measured by the LV scheme with the LV elemental analysis scaling factors applied (i.e., LV-OOA





from CalNex, BBOA from Tibet, OOA-2 from Beijing summer and LO-OOA from Beijing winter) were still lower than those measured by the TV scheme after considering the uncertainties of I-A method. In addition, some H:C ratios of OOA coatings were higher than the OOA measured by the TV scheme. When the I-A$_{sp}$ method was used, the O:C ratios generally increased compared to those determined by the I-A method with the inter-conversion factors applied

(Figures 6b and S4). In particular, LO-OOA from CalNex gave the largest enhancement of O:C ratio, followed by OOA-2 from Fontana and SV-OOA from CalNex. The H:C ratios remained roughly the same between the two methods. Combining the results presented in Figure 6a and 6b, it can be concluded that the OOA materials associated with rBC particles were likely less oxygenated (i.e., lower O:C) compared to the total OOA measured by the TV scheme at some of the sampling locations. This observation may be due to the fact that POA and/or OOA precursors

originating from combustion processes are less and/or non-oxygenated in nature. For example, our comparison shows the largest differences of O:C ratios between the two vaporization schemes for the observations from Beijing (OOA-2 in winter and LO-OOA in summer), where the air quality was expected to be significantly influenced by local combustion sources. The O:C ratio of an aged/transported BBOA factor detected by the LV scheme of SP-AMS in Tibet is also noticeably lower than that of MO-OOA detected by the HR-ToF-AMS even though they are strongly

correlated (R = 0.96). Overall, our observations indicate that even though the time series of OOA factors determined by the TV and LV scheme are strongly correlated (e.g., R > 0.9), suggesting that they were likely co-emitted or formed through similar aging processes during transport, they might contain multiple types of OA materials and their relative distribution between rBC and non-BC particles might be significantly different.

**4.    Summary**

Elemental ratios, in particularly H:C and O:C ratios, have been widely used to investigate the chemical properties of OA such as particle viscosity (Chen et al., 2011; Zhang et al., 2015), particle phase transition (Pye et al., 2017), aromatic structure or sorption properties (Xiao et al., 2016), light absorption properties (Kumar et al., 2018), and hygroscopicity (Massoli et al., 2010) in many field and laboratory studies. Enhancing the accuracy of elemental

analysis of OA is important to improve understanding of their physio-chemical properties and aging mechanisms. Although the I-A method has been widely utilized to quantify H:C and O:C ratios of OA measured by standard HR-ToF-AMS (i.e., TV scheme), the applicability of the I-A method for the elemental analysis of organic coatings that are measured by the LV scheme of SP-AMS remains uncertain, especially for ambient OOA components that are always referred to as freshly formed and aged SOA materials based on their degree of oxygenation.


To address this knowledge gap, this work examined 30 oxygenated organic species with different functional moieties, which were characterized by both TV and LV schemes of three SP-AMS instruments operated in different laboratories. The results demonstrate that the LV scheme can retain larger fragment ions during OA fragmentation compared to the TV scheme. Changes in OA fragmentation due to the LV scheme can significantly impact the $f_{C_2H_3O^+}/f_{CO_2^+}$ ratio and





$f_{C_2H_4O_2^+}$ of organic mass spectra. Therefore, the application of the observational-based framework developed based on these three organic fragments may not be straightforward for evaluating the chemical characteristics and aging of SOA and BBOA materials coated on ambient BC particles (Cubison et al., 2011; Ng et al., 2010). The I-A method is robust for determining elemental compositions of OOA materials detected by both TV and LV schemes, and the LV

elemental analysis scaling factors of 1.10 and 0.89 for H:C and O:C ratios, respectively, were determined to further improve the accuracy. The I-A$_{sp}$ method is developed in this work based on the updated multilinear regression model for the LV scheme measurements. Compared to the I-A method, the I-A$_{sp}$ method can further reduce the relative errors of O:C ratio were from -26.3% to 5.8% on average for our tested species, and the average of relative errors for H:C ratios remain roughly the same. Nevertheless, it is worth noting that the I-A$_{sp}$ method may overestimate the O:C ratio

of alcohol species, and lead to more scattered H:C ratios for dicarboxylic acids and multifunctional species. Applying the LV elemental analysis scaling factors and the I-A$_{sp}$ method to ambient data, this work demonstrates that the formation mechanisms and chemical characteristics of OOA coatings on BC particles can be different than OOA materials externally mixed with BC at the same location.

**Data availability:**

The data set for this publication is available upon contacting the corresponding authors.

**Author contributions:**

A.K.Y.L. supervised the project. Y.X.C., R.W., M.D.W., and M.R.C. carried out the experiments. M.M., L.-H.R., R.W., M.D.W., and M.R.C. analyzed the lab data. M. M., J.W., and X.G. analyzed the field data. M.M. and A.K.Y.L. wrote the manuscript with support and comments from all the co-authors.

**Competing interests:**

The authors declare that they have no conflict of interest.

**Acknowledgment:**

The authors would like to thank Max G. Adam for his assistance on initial stage of the experiments conducted at the National University of Singapore.

**Financial support:**





This work is supported by the National Environmental Agency (NEA) of Singapore (NEA, R-706-000-043-490). The content does not represent NEA's view. Measurements conducted at the University of Toronto were supported by the Natural Science and Engineering Research Council (NSERC) of Canada.



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





**Tables and Figures:**

**Table 1.** Summary of the true values of H:C, O:C and $OS_c$ of the oxygenated organic species tested by the three SP-AMS.

| Class | Name | Formula | H:C | O:C | $OS_c$ | SP-AMS |
|---|---|---|---|---|---|---|
| Multifunctional | Cis-Pinonic acid | $C_{10}H_{16}O_3$ | 1.60 | 0.30 | -1.00 | 2, 3 |
| | Citric acid | $C_6H_8O_7$ | 1.33 | 1.17 | 1.00 | 1, 2, 3 |
| | Glutamic acid | $C_5H_9NO_4$ | 1.80 | 0.80 | -0.20 | 2 |
| | Glycolic acid | $C_2H_4O_3$ | 2.00 | 1.50 | 1.00 | 1, 2 |
| | Ketoglutaric acid | $C_5H_6O_5$ | 1.20 | 1.00 | 0.80 | 3 |
| | Ketopimelic acid | $C_7H_{10}O_5$ | 1.43 | 0.71 | 0 | 3 |
| | Levulinic acid | $C_5H_8O_3$ | 1.60 | 0.60 | -0.40 | 2 |
| | Malic acid | $C_4H_6O_5$ | 1.50 | 1.25 | 1.00 | 1, 2 |
| | Pyruvic acid | $C_3H_4O_3$ | 1.33 | 1.00 | 0.67 | 2 |
| | Tartaric acid | $C_4H_6O_6$ | 1.50 | 1.50 | 1.50 | 1, 2, 3 |
| Diacids | Adipic acid | $C_6H_{10}O_4$ | 1.67 | 0.67 | -0.33 | 1, 2 |
| | Azelaic acid | $C_9H_{16}O_4$ | 1.78 | 0.44 | -0.89 | 1, 2, 3 |
| | Glutaric acid | $C_5H_8O_4$ | 1.60 | 0.80 | 0 | 1, 2, 3 |
| | Maleic Acid | $C_4H_4O_4$ | 1.00 | 1.00 | 1.00 | 2 |
| | Malonic acid | $C_3H_4O_4$ | 1.33 | 1.33 | 1.33 | 1, 2, 3 |
| | Oxalic acid | $C_2H_2O_4$ | 1.00 | 2.00 | 3.00 | 1, 2 |
| | Phthalic acid | $C_8H_6O_4$ | 0.75 | 0.50 | 0.25 | 1 |
| | Pimelic acid | $C_7H_{12}O_4$ | 1.71 | 0.57 | -0.57 | 1, 3 |
| | Suberic acid | $C_8H_{14}O_4$ | 1.75 | 0.50 | -0.75 | 1 |
| | Succinic acid | $C_4H_6O_4$ | 1.50 | 1.00 | 0.50 | 1, 2, 3 |
| Polyacids | Tricarballylic Acid | $C_6H_8O_6$ | 1.33 | 1.00 | 0.67 | 2 |
| Alcohols | Arabitol | $C_5H_{12}O_5$ | 2.40 | 1.00 | -0.40 | 1 |
| | Phenol | $C_6H_6O$ | 1.00 | 0.17 | -0.67 | 2 |
| | Xylitol | $C_5H_{12}O_5$ | 2.40 | 1.00 | -0.40 | 1, 3 |
| | 1,5-Pentanediol | $C_5H_{12}O_2$ | 2.40 | 0.40 | -1.60 | 2 |
| | Dextrose | $C_6H_{12}O_6$ | 2.00 | 1.00 | 0 | 2 |
| | Glucose | $C_6H_{12}O_6$ | 2.00 | 1.00 | 0 | 1 |
| | Sucrose | $C_{12}H_{22}O_{11}$ | 1.83 | 0.92 | 0 | 1, 3 |
| | Levoglucosan | $C_6H_{10}O_5$ | 1.67 | 0.83 | 0 | 1 |
| Esters | Bis(2-ethylhexyl) Sebacate | $C_{26}H_{50}O_4$ | 1.92 | 0.15 | -1.62 | 2 |



**Table 2**: Relative errors of H:C, O:C and $OS_c$ measured by the three independent SP-AMS

| Vaporization scheme | SP-AMS | Elemental analysis method | Average relative errors (%) | | |
|---|---|---|---|---|---|
| | | | H:C | O:C | $OS_c$ |
| Laser vaporization | 1 | I-A | 8.5% | −22.0 % | −33.2% |
| | 2 | I-A | 11.5% | −23.3 % | −18.3% |
| | 3 | I-A | −3.4% | −37.1 % | −36.9% |
| | 1, 2 & 3 | I-A | 6.6% | −26.3 % | −27.2% |
| | 1 & 2 | I-A$_{sp}$ | 6.3% | 5.8% | -9.8% |
| Thermal vaporization | 1 | I-A | −0.6 % | −21.5 % | −46.7% |
| | 2 | I-A | −5.2 % | −17.8 % | −49.5% |
| | 3 | I-A | −12.0 % | −27.0 % | −32.0% |
| | 1, 2 & 3 | I-A | −5.2 % | −21.5 % | −44.3% |



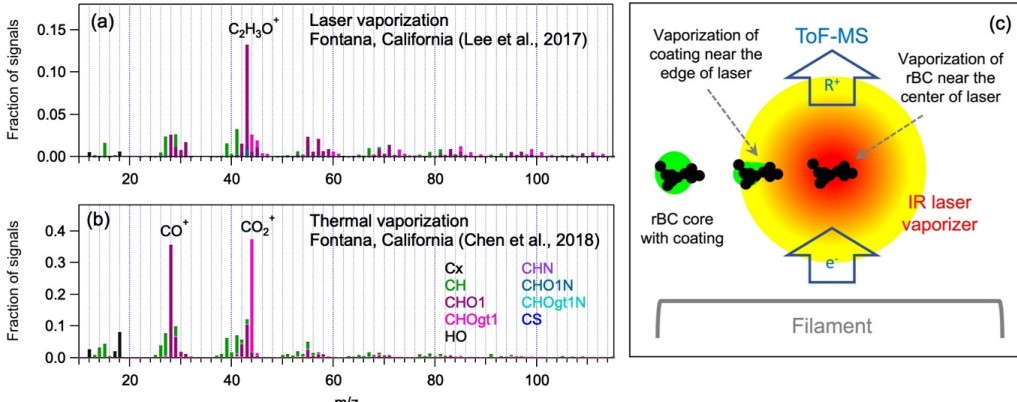

**Fig. 1**. Mass spectra of more-oxidized OOA materials measured in Fontana, California, using the (a) laser and (b) thermal vaporization schemes. (c) A simplified diagram of laser vaporizer scheme in SP-AMS. The laser intensity is stronger at the center, in which rBC can be completely vaporized at ~4000K. Organic coatings start to vaporize at the edge of laser beam during the heating process of rBC.



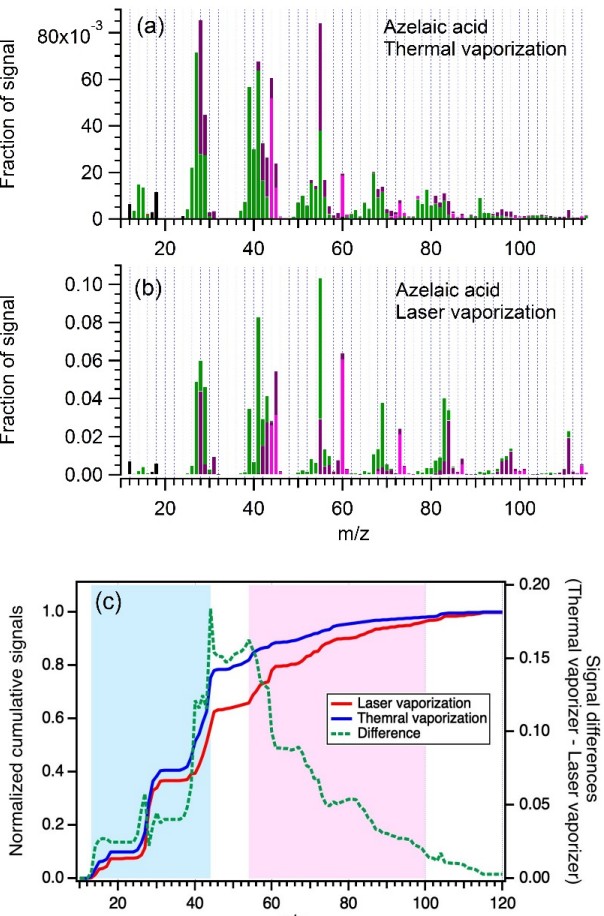

**Fig. 2.** Mass spectra of azelaic acid, measured by SP-AMS 1 using the thermal (a) and laser (b) vaporization schemes. (c) Normalized cumulative histogram of mass-to-charge ratios for the oxygenated organic compounds measured by the SP-AMS 1. The blue area indicates that the thermal vaporization scheme tends to provide organic fragments with smaller m/z, whereas the red area indicates that the laser vaporization scheme tends to give organic fragments with larger m/z.





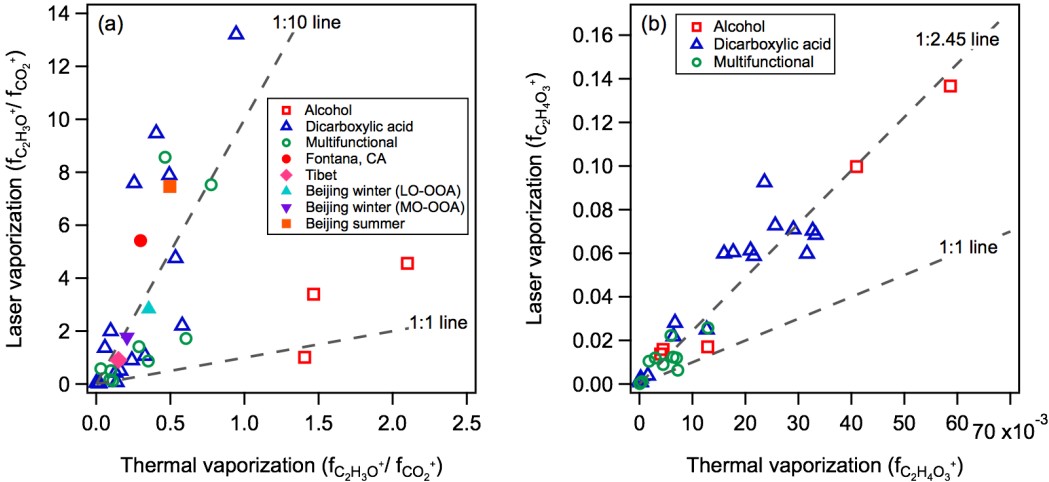

**Fig. 3.** Comparison of (a) $f_{C_2H_3O^+}/f_{CO_2^+}$ ratios and (b) $f_{C_2H_4O_2^+}$ measured by the LV and TV schemes from laboratory data (hollow markers) and collocated field measurement (solid markers) (see Table S1 for the detail of ambient data comparison).



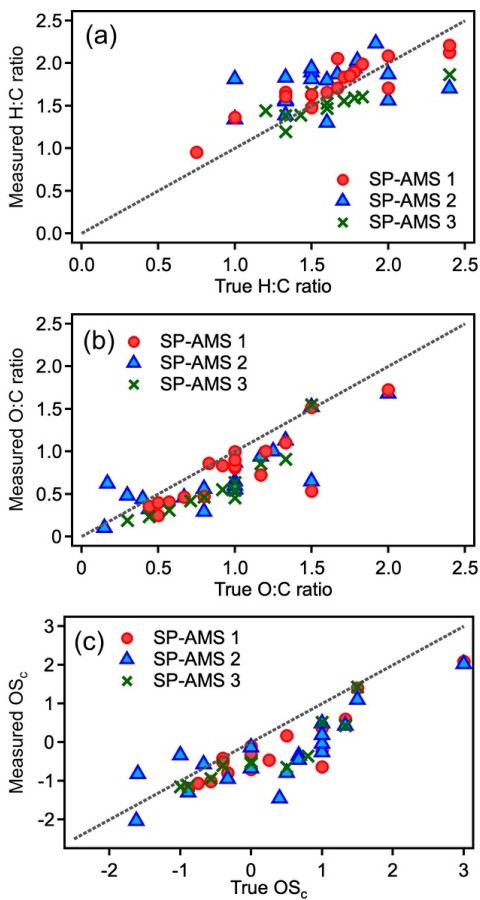

**Fig. 4.** Comparisons between the measured and true values of H:C, O:C, and $OS_c$ determined by the three SP-AMS using the LV scheme. The I-A method was used for the elemental analysis. Red circles, blue triangles, and green crosses represent data measured by SP-AMS 1, 2 and 3, respectively. The dashed lines represent 1:1 line.





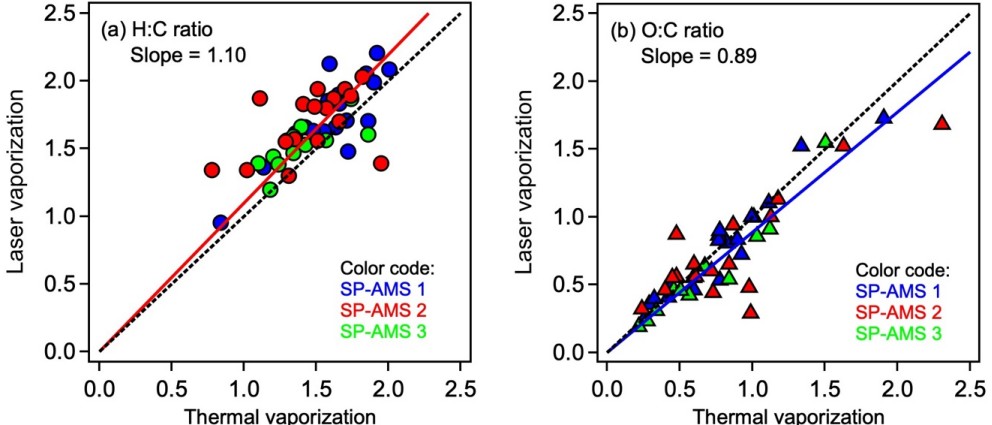

**Fig. 5.** Comparisons of H:C and O:C ratios of oxygenated organic compounds determined by the LV and TV schemes. The I-A method was used for the elemental analysis. The dashed and solid lines represent 1:1 line and data fitting, respectively.

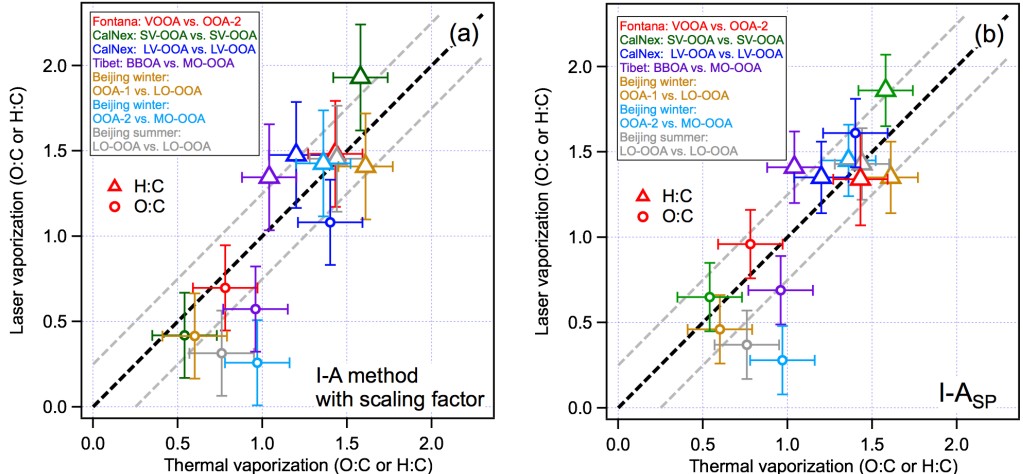

**Fig. 6.** Comparisons between ambient OOA data measured by co-located SP-AMS and HR-ToF-AMS in Nexus of

5    Air Quality and Climate Change (CalNex) 2010 campaign (Massoli et al., 2015), Fontana, California in 2015 (Chen

et al., 2018; Lee et al., 2017), Tibet in 2015 (Wang et al., 2017; Xu et al., 2018), and Beijing in 2016 (Wang et al.,

2019; Xie et al., 2019b) and 2017 (Xie et al., 2019a; Xu et al., 2019) (see Table S1 for the detail of ambient data

comparison). For panel a, the H:C and O:C ratios determined by the laser vaporization approached were corrected by

the LV elemental analysis scaling factors (1.10 for H:C and 0.89 for O:C). For panel b, the H:C and O:C ratios were

10    determined by I-A$_{sp}$ method. The LV scheme of SP-AMS were used to detect rBC-containing particles exclusively.

The error bar represents the average absolute errors of I-A (TV in panels a and b, and LV in panel a) and I-A$_{sp}$ (LV in

panel b) for determining the H:C and O:C ratios. The dashed black line represents the 1:1 line and the dashed grey

line indicates ±0.25 of l:1 line.