# Peer review of "Elemental analysis of Oxygenated Organic Coating on Black Carbon Particles using a Soot-Particle Aerosol Mass Spectrometer"

_Atmospheric Measurement Techniques, 2020_

## Referee Comment (RC1) · Anonymous Referee #1 · 8 Sep 2020

Ma et al., reported the response of oxygenated organics in the SP laser in the AMS and found that comparing to using a standard tungsten vaporizer, the SP mode generates larger ion fragments. Using laboratory-generated pure organic aerosols, they developed an SP-based improved-ambient method for estimating aerosol elemental ratio. This manuscript addresses an important topic in AMS application, and is within the scope of the journal. I recommend publishing the paper after the authors address the following concerns.

Major comments:

1. How are RIEs of organics affected by LV vs TV? Is it possible that the lower degree

of fragmentation in LV (thus larger molecular-size species) will generate fragments with lower molecular speed in the ionization region and lead to an increase in RIE?

2. In Figure 4b, some data points show measured O:C that are almost 3 times lower than the true O:C. What are those compounds and what causes such big discrepancies?

3. How would LV vs TV affect the fragmentation of other coating species such as nitrate and more reduced hydrocarbons (e.g. f57)? The authors mentioned that organic coating on BC core appear to be less oxygenated compared to those externally mixed with BC possibly due to the co-emitted POA. It will be important to know if representative POA fragments are different in laser vaporization vs thermal vaporization. In addition, it is important to understand how nitrate fragments (both inorganic and organic nitrates) change in laser vaporization. The difference in NO:NO2 ratio may affect organic nitrate quantification.

Minor comments:

1. Line 11: What is "*their* atmospheric transport and lifetime" referred to? organic coatings? BC particles? Or cloud formation potential? 2. Line 27" A total of . . ., were included. . ."
* * *

---

## Referee Comment (RC2) · Anonymous Referee #2 · 24 Sep 2020

The manuscript by Ma et al. entitled 'Elemental analysis of Oxygenated Organic Coating on Black Carbon Particles using a Soot-Particle Aerosol Mass Spectrometer' focuses on validation of elemental ratio analysis by the soot-particle aerosol mass spectrometer (SP-AMS). The approach includes both laboratory and field studies. Three SP-AMSs were employed for the study, allowing inter-comparison of the data. Chemical characteristics of coating material on soot particles is important for understanding the climatic impacts of aerosol particles. The content of the manuscript is within the focus of the journal. The manuscript is well written, and easy to follow. I support publication of the manuscript after addressing the following comments.

[Figure]

1. Morphology of particles for laboratory study. The authors mixed Regal Black and aqueous solution of standard organic compounds. Subsequently, particles were generated using an atomizer. I wonder how the morphology of these particles are. In the case of ambient 'coated soot particles,' coating material should be located on the surface of particles. However, the generation method could produce particles with different morphology (e.g., homogeneously mixed; light-absorbing material is coating non-absorbing material). The discussion would be more convincing if the authors could describe the potential influence of particle morphology on experimental data.

2. Stability of Regal Black mass spectra The contributions of Regal Black on observed mass spectra were subtracted using a fragmentation table. I wonder if there were any differences in mass spectra of Regal Black those were observed by different instruments.

---

## Referee Comment (RC3) · Anonymous Referee #3 · 3 Nov 2020

The manuscript "Elemental analysis of Oxygenated Organic Coating on Black Carbon Particles using a Soot-Particle Aerosol Mass Spectrometer" by Mutian Ma et al. presents fundamental work on the use of electron-ionization mass spectra for estimates of elemental analysis (EA) organic compounds. The manuscript focusses on the differences between the laser vaporizer in the SP-AMS and the thermal vaporizer in the predecessor AMS. The main goal of the manuscript is to present a revised parameterization for EA estimation.

The work is excellent, and the presentation is outstandingly clear. I have a few short comments on the statistical presentation of the results which the authors should be

able to address easily. I recommend publication after these minor comments.

First, the major result of this manuscript is the I-A,SP parameterization from laboratory work. The graph which truly illustrates this result is Figure S3. Figure 4 shows the "old" method. I recommend that the authors combine Figures 4 and S3 into one 6 panel figure.

Second, the abstract discusses the "relative error of O:C" for the compounds measured in the lab. The manuscript explains that this is the "average relative error". I am not entirely clear how the average was calculated, but I believe this is the Root Mean Squared Error (RMSE) or Mean Absolute Error (MAE). The authors should specify this precisely.

The RMSE or MAE is a measure of the bias of the calibration. The authors should also report the precision of the calibration. An excellent example of this is found in Reggente, Dillner, and Takahama (Atmos Meas Tech 2016, https://doi.org/10.5194/amt-9-441-2016), but the authors may prefer some other formulation. My first comment also addresses the importance of precision, because the scatter in Figures S3 and 4 shows this precision.

---

## Author Comment (AC1) · 16 Feb 2021

**Responses to the reviewers:**

**"Elemental analysis of Oxygenated Organic Coating on Black Carbon Particles using a Soot-Particle Aerosol Mass Spectrometer" by Mutian Ma et al.**

**Reviewer #1**

Ma et al., reported the response of oxygenated organics in the SP laser in the AMS and found that comparing to using a standard tungsten vaporizer, the SP mode generates larger ion fragments. Using laboratory-generated pure organic aerosols, they developed an SP-based improved-ambient method for estimating aerosol elemental ratio. This manuscript addresses an important topic in AMS application, and is within the scope of the journal. I recommend publishing the paper after the authors address the following concerns.

Response: We thank for the positive feedback from the reviewer on the significance of this work. Our responses to specific comments are shown in blue color below:

Major comments:

1. How are RIEs of organics affected by LV vs TV? Is it possible that the lower degree of fragmentation in LV (thus larger molecular-size species) will generate fragments with lower molecular speed in the ionization region and lead to an increase in RIE?

   Response: Thanks for addressing this important issue. Recent field and laboratory studies have shown that the dual vaporization (DV) mode (i.e., LV is operated in the presence of TV) leads to higher mass loadings of organics (and other components of non-refractory particulate matter (NR-PM)) compared to those observed by the TV scheme when the ionization efficiency (IE) of nitrate (i.e., a calibration standard for quantifying NR-PM) determined by TV mode was used in the data analysis (Ahern et al., 2016; Avery et al., 2020; Lee et al., 2015). Such observation suggests that 1) the LV scheme may result in higher IE (and/or RIE) of organics and/or 2) the LV scheme can provide higher collection efficiencies (CE) for particles that cannot be captured by the TV scheme. However, the relative contribution of these two possibilities remain unclear, and we do not have sufficient information from our work to comment if the degree of fragmentation can be the main contributor to the IE/RIE enhancements for coatings on BC particles.

   Although the changes in IE or/and RIE due to the LV scheme is not the focus of this work, this is certainly a technical challenge that should be solved in the near future within the SP-AMS community. There are some on-going research within the SP-AMS community to improve our fundamental understanding on this topic. For examples, a recent work by Avery et al. (2020) investigated the effect of the laser-heating interference, reporting that laser heating only have a minor effect on the NR-PM ion signals. Single particle-based IE/RIE calibration method for quantifying organic and other NR-PM coating materials on BC particles is currently under investigation in our laboratory. More recent discussion regarding this topic can be found in the 21[st] AMS users meeting hold on January 2021 (http://tinyurl.com/AMS-Users).

2. In Figure 4b, some data points show measured O:C that are almost 3 times lower than the true O:C. What are those compounds and what causes such big discrepancies?

   Response: The I-A method (Canagaratna et al., 2015) improves the accuracy with which elemental ratios are calculated compared to the A-A method (Aiken et al., 2007) by including a more complete parameterization that utilizes measured $CO_2^+$ and $CHO^+$ signal mass fractions ($f_{CHO^+}$ and $f_{CO_2^+}$). As pointed out by Canagaratna et al. (2015), $f_{CHO^+}$ and $f_{CO_2^+}$ can vary between different organic acids and alcohols, and a single I-A parameterization cannot capture all individual species with the same accuracy. In this work, the species with the underpredicted O:C values are glycolic acid and glutamic acid. Both species have low $CHO^+$ and $CO_2^+$ signals, contributing only 1-2% of total organic mass. The following sentence has been added to highlight this limitation in Section 3.4 of the revised manuscript as shown below:

Section 3.4, page 7, lines 29-31: "Note that some organic species, such as glycolic acid and glutamic acid, give relatively large discrepancies between the measured and the true O:C values that can be due to their low contribution of $CHO^+$ and $CO_2^+$ signals to the total organic mass (e.g.,1-2%)."

3. How would LV vs TV affect the fragmentation of other coating species such as nitrate and more reduced hydrocarbons (e.g. f57)? The authors mentioned that organic coating on BC core appear to be less oxygenated compared to those externally mixed with BC possibly due to the co-emitted POA. It will be important to know if representative POA fragments are different in laser vaporization vs thermal vaporization. In addition, it is important to understand how nitrate fragments (both inorganic and organic nitrates) change in laser vaporization. The difference in NO:NO2 ratio may affect organic nitrate quantification.

Response: The primary focus of this work is to investigate the effects of vaporization scheme on fragmentation of oxygenated organic aerosol (OOA) species that can be found in ambient SOA and BBOA materials. For the OOA species tested in this work, although the average enhancement factor for f57 of ~1.8 was observed, f57 only have minimal contributions to total organic mass of each species (i.e. <1%). A few previous studies have shown that there were some mass spectral differences (e.g., $C_3H_5^+/C_3H_7^+$ and $C_5H_7^+/C_5H_9^+$ ratios) between the HOA factors identified by the TV and LV modes but their overall mass spectra are still more comparable to each other compared to the OOA factors (Chen et al., 2018; Lee et al., 2017; Massoli et al., 2015; Wang et al., 2020). Nevertheless, it is important to note that including rBC fragment in PMF analysis of LV mode data can generate additional traffic/combustion related POA factor that is more associated with rBC in some urban studies (Decesari et al., 2014; Lee et al., 2017; Rivellini et al., 2020; Wang et al., 2020). This BC-rich POA factor has organic mass spectrum has some difference from the HOA factor identified by both LV and TV modes at the same location (Chen et al., 2018; Decesari et al., 2014; Lee et al., 2017; Lee et al., 2019b). Cooking-related OA (COA) is another major POA component but it is likely largely externally mixed with BC based on previous urban observations (Chen et al., 2018; He et al., 2019; Lee et al., 2017; Lee et al., 2015; Rivellini et al., 2020; Wang et al., 2018). Some of the above information has been added in the introduction as shown in the first modifications below. The second modification has been made to clearly highlight the focus of this work:

Introduction, page 2, lines 29-34 "Previous field studies have also shown that there were some mass spectral differences (e.g., $C_3H_5^+/C_3H_7^+$ and $C_5H_7^+/C_5H_9^+$ ratios) between the hydrocarbon-like OA (HOA) factors, which usually represents primary OA (POA) components emitted from traffic/fossil fuel combustion processes, identified by the TV and LV schemes at the same location (Chen et al., 2018; Lee et al., 2017; Massoli et al., 2015; Wang et al., 2020; Wang et al., 2017; Wang et al., 2019; Xie et al., 2019a; Xie et al., 2019b)."

Introduction, page 3, line 17-18: "This work focuses on investigating oxygenated organic compounds as significant mass spectral differences were observed from ambient OOA components detected by TV and LV schemes."

Regarding our argument in Section 3.6, we are referring to those ambient OA particles that had experienced enough aging to be interpreted as OOA based on their mass spectral features and elemental compositions (e.g. O:C). However, in some heavily polluted urban environments (e.g., Beijing in this work), OOA can be largely influenced by local atmospheric processing of primary emissions and they can be formed through 1) gas-phase oxidation of anthropogenic VOCs and 2) heterogeneous processing of POA (such as HOA and COA). Therefore, the actual meaning of our argument is that OOA precursors emitted from combustion processes, such as POA, are non- and relatively less oxygenated in nature and they are likely co-emitted and internally mixed with BC. If POA is one of the major starting materials for generating those OOA coated on BC in urban environments, their degree of oxygenation may be lower than those formed via gas-phase oxidation of VOCs as gas-phase products have to reach higher level of oxygenation (and thus volatility) to condense as OOA. Also, condensation of those gas-phase OOA can occur onto any types of existing particles and is not selective to BC particles. Note that our previous study demonstrated that BC is unlikely to be the major condensation sinks of OOA produced near local traffic emissions (Lee et al., 2017). To describe this concept clearly, the related sentence in Section 3.6 has been revised as following:

Section 3.6, page 10, lines 29-30: "This observation may be due to the fact that OOA formed from heterogeneous oxidation of POA such as HOA that is co-emitted and internally mixed with BC is likely to be less and/or non-oxygenated in nature."

Lastly, the authors understand that organo-nitrate can significantly contribute to the total OA mass in some locations, and hence it is also important to better understand the changes in fragmentation of nitrate due to different vaporization scheme. Although we did not conduct experiments for organo-nitrate species in this work, we further investigate whether any differences of inorganic nitrate fragmentation (i.e., ratio of $NO^+$ and $NO_2^+$) between the TV and LV modes using pure ammonium nitrate particles. Our observations show that $NO^+/NO_2^+$ ratios measured by the TV and LV schemes are 1.9 and 0.7, respectively. This ratio of 1.9 obtained from TV is comparable to 1.7-2.2 reported by Xu et al. (2015). Since the nitrate fragmentation pattern can make significant impact on quantification of particulate organo-nitrate in the atmosphere (Farmer et al., 2010; Fry et al., 2009; Xu et al., 2015), the above information has been added to the revised manuscript as shown below.

Section 3.1, page 5, line 32 - page 6, line 2, "It is worth mentioning that although organo-nitrate compounds, which have significant contribution to SOA mass in some locations (Fry et al., 2018; Lee et al., 2019a; Xu et al., 2015; Xu et al., 2017), were not tested in this work, changes in fragmentation for ammonium nitrate (i.e., AMS calibration standard) due to vaporization scheme were observed. Our observations show that the $NO^+/NO_2^+$ ratios of ammonium nitrate measured by the TV and LV schemes are equal to 1.9 and 0.7, respectively, suggesting that the quantification of organo-nitrate compounds based on the $NO^+/NO_2^+$ ratios approach can also be affected by the vaporization scheme (Farmer et al., 2010; Xu et al., 2015)."

Conclusions, page 11, line 31 – page 12 line 2: "Lastly, a significant difference between the $NO^+/NO_2^+$ ratios for ammonium nitrate particles measured by the TV and LV scheme was observed in this work. As the $NO^+/NO_2^+$ ratios have been widely used for quantifying particle-phase organo-nitrate compounds, this observation suggests that changes in fragmentation can have an impact on quantifying organo-nitrate compounds coated on BC."

Minor comments:

1. Line 11: What is "*their* atmospheric transport and lifetime" referred to? organic coatings? BC particles? Or cloud formation potential? 2. Line 27" A total of . . ., were included. . ."

   Responses:

   The manuscript has been revised as shown below:

   Page 2, Line 11: It is referred to BC particles. The sentence has been changed to "…subsequently atmospheric transport and lifetime of BC particles".

   Page 4, Line 3: This grammatical error has been corrected as suggested.

**Reference**

Ahern, A. T., Subramanian, R., Saliba, G., Lipsky, E. M., Donahue, N. M., and Sullivan, R. C.: Effect of secondary organic aerosol coating thickness on the real-time detection and characterization of biomass-burning soot by two particle mass spectrometers, Atmos. Meas. Tech., 9, 6117-6137, 2016.

Aiken, A. C., DeCarlo, P. F., and Jimenez, J. L.: Elemental analysis of organic species with electron ionization high-resolution mass spectrometry, Anal. Chem., 79, 8350-8358, 2007.

Avery, A. M., Williams, L. R., Fortner, E. C., Robinson, W. A., and Onasch, T. B.: Particle detection using the dual-vaporizer configuration of the Soot Particle Aerosol Mass Spectrometer (SP-AMS), Aerosol Sci. Tech., doi: 10.1080/02786826.2020.1844132, 2020. 1-15, 2020.

Canagaratna, M. R., Jimenez, J. L., Kroll, J. H., Chen, Q., Kessler, S. H., Massoli, P., Hildebrandt Ruiz, L., Fortner, E., Williams, L. R., Wilson, K. R., Surratt, J. D., Donahue, N. M., Jayne, J. T., and Worsnop, D. R.: Elemental ratio measurements of organic compounds using aerosol mass spectrometry: characterization, improved calibration, and implications, Atmos. Chem. Phys., 15, 253-272, 2015.

Chen, C. L., Chen, S., Russell, L. M., Liu, J., Price, D. J., Betha, R., Sanchez, K. J., Lee, A. K. Y., Williams, L., Collier, S. C., Zhang, Q., Kumar, A., Kleeman, M. J., Zhang, X., and Cappa, C. D.: Organic Aerosol Particle Chemical Properties Associated With Residential Burning and Fog in Wintertime San Joaquin Valley (Fresno) and With Vehicle and Firework Emissions in Summertime South Coast Air Basin (Fontana), J. Geophys. Res.-Atmos., 123, 10,707-710,731, 2018.

Decesari, S., Allan, J., Plass-Duelmer, C., Williams, B. J., Paglione, M., Facchini, M. C., O'Dowd, C., Harrison, R. M., Gietl, J. K., Coe, H., Giulianelli, L., Gobbi, G. P., Lanconelli, C., Carbone, C., Worsnop, D., Lambe, A. T., Ahern, A. T., Moretti, F., Tagliavini, E., Elste, T., Gilge, S., Zhang, Y., and Dall'Osto, M.: Measurements of the aerosol chemical composition and mixing state in the Po Valley using multiple spectroscopic techniques, Atmos. Chem. Phys., 14, 12109-12132, 2014.

Farmer, D. K., Matsunaga, A., Docherty, K. S., Surratt, J. D., Seinfeld, J. H., Ziemann, P. J., and Jimenez, J. L.: Response of an aerosol mass spectrometer to organonitrates and organosulfates and implications for atmospheric chemistry, Proc. Natl. Acad. Sci. U S A, 107, 6670-6675, 2010.

Fry, J. L., Brown, S. S., Middlebrook, A. M., Edwards, P. M., Campuzano-Jost, P., Day, D. A., Jimenez, J. L., Allen, H. M., Ryerson, T. B., Pollack, I., Graus, M., Warneke, C., de Gouw, J. A., Brock, C. A., Gilman, J., Lerner, B. M., Dubé, W. P., Liao, J., and Welti, A.: Secondary organic aerosol (SOA) yields from $NO_3$ radical + isoprene based on nighttime aircraft power plant plume transects, Atmos. Chem. Phys., 18, 11663-11682, 2018.

Fry, J. L., Kiendler-Scharr, A., Rollins, A. W., Wooldridge, P. J., Brown, S. S., Fuchs, H., Dubé, W., Mensah, A., dal Maso, M., Tillmann, R., Dorn, H. P., Brauers, T., and Cohen, R. C.: Organic nitrate and secondary organic aerosol yield from $NO_3$ oxidation of β-pinene evaluated using a gas-phase kinetics/aerosol partitioning model, Atmos. Chem. Phys., 9, 1431-1449, 2009.

He, Y., Sun, Y., Wang, Q., Zhou, W., Xu, W., Zhang, Y., Xie, C., Zhao, J., Du, W., Qiu, Y., Lei, L., Fu, P., Wang, Z., and Worsnop, D. R.: A Black Carbon-Tracer Method for Estimating Cooking Organic Aerosol From Aerosol Mass Spectrometer Measurements, Geophys. Res. Lett., 46, 8474-8483, 2019.

Lee, A. K. Y., Adam, M. G., Liggio, J., Li, S.-M., Li, K., Willis, M. D., Abbatt, J. P. D., Tokarek, T. W., Odame-Ankrah, C. A., Osthoff, H. D., Strawbridge, K., and Brook, J. R.: A large contribution of anthropogenic organo-nitrates to secondary organic aerosol in the Alberta oil sands, Atmos. Chem. Phys., 19, 12209-12219, 2019a.

Lee, A. K. Y., Chen, C.-L., Liu, J., Price, D. J., Betha, R., Russell, L. M., Zhang, X., and Cappa, C. D.: Formation of secondary organic aerosol coating on black carbon particles near vehicular emissions, Atmos. Chem. Phys., 17, 15055-15067, 2017.

Lee, A. K. Y., Rivellini, L. H., Chen, C. L., Liu, J., Price, D. J., Betha, R., Russell, L. M., Zhang, X., and Cappa, C. D.: Influences of Primary Emission and Secondary Coating Formation on the Particle Diversity and Mixing State of Black Carbon Particles, Environ. Sci. Technol., 53, 9429-9438, 2019b.

Lee, A. K. Y., Willis, M. D., Healy, R. M., Onasch, T. B., and Abbatt, J. P. D.: Mixing state of carbonaceous aerosol in an urban environment: single particle characterization using the soot particle aerosol mass spectrometer (SP-AMS), Atmos. Chem. Phys., 15, 1823-1841, 2015.

Massoli, P., Onasch, T. B., Cappa, C. D., Nuamaan, I., Hakala, J., Hayden, K., Li, S.-M., Sueper, D. T., Bates, T. S., Quinn, P. K., Jayne, J. T., and Worsnop, D. R.: Characterization of black carbon-containing particles from soot particle aerosol mass spectrometer measurements on the R/VAtlantisduring CalNex 2010, J. Geophys. Res.-Atmos., 120, 2575-2593, 2015.

Rivellini, L. H., Adam, M. G., Kasthuriarachchi, N., and Lee, A. K. Y.: Characterization of carbonaceous aerosols in Singapore: insight from black carbon fragments and trace metal ions detected by a soot particle aerosol mass spectrometer, Atmos. Chem. Phys., 20, 5977-5993, 2020.

Wang, J., Ye, J., Liu, D., Wu, Y., Zhao, J., Xu, W., Xie, C., Shen, F., Zhang, J., Ohno, P. E., Qin, Y., Zhao, X., Martin, S. T., Lee, A. K. Y., Fu, P., Jacob, D. J., Zhang, Q., Sun, Y., Chen, M., and Ge, X.: Characterization of submicron organic particles in Beijing during summertime: comparison between SP-AMS and HR-AMS, Atmos. Chem. Phys., 20, 14091-14102, 2020.

Wang, J., Zhang, Q., Chen, M., Collier, S., Zhou, S., Ge, X., Xu, J., Shi, J., Xie, C., Hu, J., Ge, S., Sun, Y., and Coe, H.: First Chemical Characterization of Refractory Black Carbon Aerosols and Associated Coatings over the Tibetan Plateau (4730 m a.s.l), Environ. Sci. Technol., 51, 14072-14082, 2017.

Wang, J. F., Liu, D. T., Ge, X. L., Wu, Y. Z., Shen, F. Z., Chen, M. D., Zhao, J., Xie, C. H., Wang, Q. Q., Xu, W. Q., Zhang, J., Hu, J. L., Allan, J., Joshi, R., Fu, P. Q., Coe, H., and Sun, Y. L.: Characterization of black carbon-containing fine particles in Beijing during wintertime, Atmospheric Chemistry and Physics, 19, 447-458, 2019.

Wang, J. F., Wu, Y. Z., Ge, X. L., Shen, Y. F., Ge, S., and Chen, M. D.: Characteristics and sources of ambient refractory black carbon aerosols: Insights from soot particle aerosol mass spectrometer, Atmos. Environ., 185, 147-152, 2018.

Xie, C., Xu, W., Wang, J., Liu, D., Ge, X., Zhang, Q., Wang, Q., Du, W., Zhao, J., Zhou, W., Li, J., Fu, P., Wang, Z., Worsnop, D., and Sun, Y.: Light absorption enhancement of black carbon in urban Beijing in summer, Atmos. Environ., 213, 499-504, 2019a.

Xie, C., Xu, W., Wang, J., Wang, Q., Liu, D., Tang, G., Chen, P., Du, W., Zhao, J., Zhang, Y., Zhou, W., Han, T., Bian, Q., Li, J., Fu, P., Wang, Z., Ge, X., Allan, J., Coe, H., and Sun, Y.: Vertical characterization of aerosol optical properties and brown carbon in winter in urban Beijing, China, Atmos. Chem. Phys., 19, 165-179, 2019b.

Xu, L., Suresh, S., Guo, H., Weber, R. J., and Ng, N. L.: Aerosol characterization over the southeastern United States using high-resolution aerosol mass spectrometry: spatial and seasonal variation of aerosol composition and sources with a focus on organic nitrates, Atmos. Chem. Phys., 15, 7307-7336, 2015.

Xu, W., Sun, Y., Wang, Q., Du, W., Zhao, J., Ge, X., Han, T., Zhang, Y., Zhou, W., Li, J., Fu, P., Wang, Z., and Worsnop, D. R.: Seasonal Characterization of Organic Nitrogen in Atmospheric Aerosols Using High Resolution Aerosol Mass Spectrometry in Beijing, China, ACS Earth and Space Chem., 1, 673-682, 2017.

---

## Author Comment (AC2) · 16 Feb 2021

**Responses to the reviewers:**

**"Elemental analysis of Oxygenated Organic Coating on Black Carbon Particles using a Soot-Particle Aerosol Mass Spectrometer" by Mutian Ma et al.**

**Reviewer #2**

The manuscript by Ma et al. entitled 'Elemental analysis of Oxygenated Organic Coating on Black Carbon Particles using a Soot-Particle Aerosol Mass Spectrometer' focuses on validation of elemental ratio analysis by the soot-particle aerosol mass spectrometer (SP-AMS). The approach includes both laboratory and field studies. Three SP-AMSs were employed for the study, allowing inter-comparison of the data. Chemical characteristics of coating material on soot particles is important for understanding the climatic impacts of aerosol particles. The content of the manuscript is within the focus of the journal. The manuscript is well written, and easy to follow. I support publication of the manuscript after addressing the following comments.

Response: We thank for the constructive comments and questions from the reviewer. Our responses to specific comments are shown in blue color below:

Specific comments:

1.  Morphology of particles for laboratory study. The authors mixed Regal Black and aqueous solution of standard organic compounds. Subsequently, particles were generated using an atomizer. I wonder how the morphology of these particles are. In the case of ambient 'coated soot particles,' coating material should be located on the surface of particles. However, the generation method could produce particles with different morphology (e.g., homogeneously mixed; light-absorbing material is coating non-absorbing material). The discussion would be more convincing if the authors could describe the potential influence of particle morphology on experimental data.

    Response: We agree with the reviewer that the laboratory-generated particles can be more spherical and homogenously mixed compared to ambient particles. It has been well known that the particle beam width of ambient BC particles strongly depends on its morphology, which can greatly affect collection efficiency of ambient BC particles due to incomplete overlapping between laser and particle beams (Willis et al., 2014). However, whether BC particle morphology can affect the fragmentation of coating materials remains poorly investigated. Since different fragmentation of organic coating observed in this work is likely due to lower vaporization temperature for those vaporized from BC particle surface, we speculate that the particle morphology may affect how fast the BC can be heated up but not the vaporization temperature of organic coating, and thus may not have significant impact on the fragmentation pattern. The general agreement of f44/f43 enhancement (i.e., the two major oxygenated fragments observed for ambient OOA) between our lab observations and field data also provide indirect evidence to support our speculation. However, we would like to highlight this uncertainty in our revised manuscript as this will require further investigation to confirm our speculation in the future as following.

    Section 3.6, page 9, line 31 – page 10 line 2: "Given that our observations that vaporization scheme plays a critical role in the fragmentation process of oxygenated organic species, the LV elemental analysis scaling factors (0.89 for H:C and 1.10 for O:C) and the I-A$_{sp}$ method obtained in this work can facilitate more direct and robust comparison between the two types of measurements based on elemental analysis. Nevertheless, it is important to note that rBC and organics were likely more homogeneously mixed within our laboratory-generated particles, which can be very different from the morphology of ambient organically coated BC particles. Therefore, such inter-instrument comparisons assume that BC morphology is not a key factor to affect organic fragmentation observed by the LV scheme.

2.  Stability of Regal Black mass spectra. The contributions of Regal Black on observed mass spectra were subtracted using a fragmentation table. I wonder if there were any differences in mass spectra of Regal Black those were observed by different instruments.

    Response: The Regal Black mass spectra can have small different between instruments. However, we observed that Regal Black mass spectra is consistently obtained from SP-AMS 1 in different calibrations. It is important to emphasized that Regal Black calibration was performed for each instrument so that the fragmentation table correction (i.e., $C_1^+$:$C_3^+$, $CO^+$:$C_3^+$ and $CO_2^+$:$C_3^+$ ratios) applied for rBC is instrument specific. This information has been added to Section 2.3 as shown below.

    Section 2.3, page 5, lines 9-10: "Regal Black calibrations were performed for each instrument and thus the applied corrections for fragmentation table were instrument specific."

**Reference**

Ahern, A. T., Subramanian, R., Saliba, G., Lipsky, E. M., Donahue, N. M., and Sullivan, R. C.: Effect of secondary organic aerosol coating thickness on the real-time detection and characterization of biomass-burning soot by two particle mass spectrometers, Atmos. Meas. Tech., 9, 6117-6137, 2016.

Aiken, A. C., DeCarlo, P. F., and Jimenez, J. L.: Elemental analysis of organic species with electron ionization high-resolution mass spectrometry, Anal. Chem., 79, 8350-8358, 2007.

Avery, A. M., Williams, L. R., Fortner, E. C., Robinson, W. A., and Onasch, T. B.: Particle detection using the dual-vaporizer configuration of the Soot Particle Aerosol Mass Spectrometer (SP-AMS), Aerosol Sci. Tech., doi: 10.1080/02786826.2020.1844132, 2020. 1-15, 2020.

Canagaratna, M. R., Jimenez, J. L., Kroll, J. H., Chen, Q., Kessler, S. H., Massoli, P., Hildebrandt Ruiz, L., Fortner, E., Williams, L. R., Wilson, K. R., Surratt, J. D., Donahue, N. M., Jayne, J. T., and Worsnop, D. R.: Elemental ratio measurements of organic compounds using aerosol mass spectrometry: characterization, improved calibration, and implications, Atmos. Chem. Phys., 15, 253-272, 2015.

Chen, C. L., Chen, S., Russell, L. M., Liu, J., Price, D. J., Betha, R., Sanchez, K. J., Lee, A. K. Y., Williams, L., Collier, S. C., Zhang, Q., Kumar, A., Kleeman, M. J., Zhang, X., and Cappa, C. D.: Organic Aerosol Particle Chemical Properties Associated With Residential Burning and Fog in Wintertime San Joaquin Valley (Fresno) and With Vehicle and Firework Emissions in Summertime South Coast Air Basin (Fontana), J. Geophys. Res.-Atmos., 123, 10,707-710,731, 2018.

Decesari, S., Allan, J., Plass-Duelmer, C., Williams, B. J., Paglione, M., Facchini, M. C., O'Dowd, C., Harrison, R. M., Gietl, J. K., Coe, H., Giulianelli, L., Gobbi, G. P., Lanconelli, C., Carbone, C., Worsnop, D., Lambe, A. T., Ahern, A. T., Moretti, F., Tagliavini, E., Elste, T., Gilge, S., Zhang, Y., and Dall'Osto, M.: Measurements of the aerosol chemical composition and mixing state in the Po Valley using multiple spectroscopic techniques, Atmos. Chem. Phys., 14, 12109-12132, 2014.

Farmer, D. K., Matsunaga, A., Docherty, K. S., Surratt, J. D., Seinfeld, J. H., Ziemann, P. J., and Jimenez, J. L.: Response of an aerosol mass spectrometer to organonitrates and organosulfates and implications for atmospheric chemistry, Proc. Natl. Acad. Sci. U S A, 107, 6670-6675, 2010.

Fry, J. L., Brown, S. S., Middlebrook, A. M., Edwards, P. M., Campuzano-Jost, P., Day, D. A., Jimenez, J. L., Allen, H. M., Ryerson, T. B., Pollack, I., Graus, M., Warneke, C., de Gouw, J. A., Brock, C. A., Gilman, J., Lerner, B. M., Dubé, W. P., Liao, J., and Welti, A.: Secondary organic aerosol (SOA) yields from $NO_3$ radical + isoprene based on nighttime aircraft power plant plume transects, Atmos. Chem. Phys., 18, 11663-11682, 2018.

Fry, J. L., Kiendler-Scharr, A., Rollins, A. W., Wooldridge, P. J., Brown, S. S., Fuchs, H., Dubé, W., Mensah, A., dal Maso, M., Tillmann, R., Dorn, H. P., Brauers, T., and Cohen, R. C.: Organic nitrate and secondary organic aerosol yield from $NO_3$ oxidation of β-pinene evaluated using a gas-phase kinetics/aerosol partitioning model, Atmos. Chem. Phys., 9, 1431-1449, 2009.

He, Y., Sun, Y., Wang, Q., Zhou, W., Xu, W., Zhang, Y., Xie, C., Zhao, J., Du, W., Qiu, Y., Lei, L., Fu, P., Wang, Z., and Worsnop, D. R.: A Black Carbon-Tracer Method for Estimating Cooking Organic Aerosol From Aerosol Mass Spectrometer Measurements, Geophys. Res. Lett., 46, 8474-8483, 2019.

Lee, A. K. Y., Adam, M. G., Liggio, J., Li, S.-M., Li, K., Willis, M. D., Abbatt, J. P. D., Tokarek, T. W., Odame-Ankrah, C. A., Osthoff, H. D., Strawbridge, K., and Brook, J. R.: A large contribution of anthropogenic organo-nitrates to secondary organic aerosol in the Alberta oil sands, Atmos. Chem. Phys., 19, 12209-12219, 2019a.

Lee, A. K. Y., Chen, C.-L., Liu, J., Price, D. J., Betha, R., Russell, L. M., Zhang, X., and Cappa, C. D.: Formation of secondary organic aerosol coating on black carbon particles near vehicular emissions, Atmos. Chem. Phys., 17, 15055-15067, 2017.

Lee, A. K. Y., Rivellini, L. H., Chen, C. L., Liu, J., Price, D. J., Betha, R., Russell, L. M., Zhang, X., and Cappa, C. D.: Influences of Primary Emission and Secondary Coating Formation on the Particle Diversity and Mixing State of Black Carbon Particles, Environ. Sci. Technol., 53, 9429-9438, 2019b.

Lee, A. K. Y., Willis, M. D., Healy, R. M., Onasch, T. B., and Abbatt, J. P. D.: Mixing state of carbonaceous aerosol in an urban environment: single particle characterization using the soot particle aerosol mass spectrometer (SP-AMS), Atmos. Chem. Phys., 15, 1823-1841, 2015.

Massoli, P., Onasch, T. B., Cappa, C. D., Nuamaan, I., Hakala, J., Hayden, K., Li, S.-M., Sueper, D. T., Bates, T. S., Quinn, P. K., Jayne, J. T., and Worsnop, D. R.: Characterization of black carbon-containing particles from soot particle aerosol mass spectrometer measurements on the R/VAtlantisduring CalNex 2010, J. Geophys. Res.-Atmos., 120, 2575-2593, 2015.

Rivellini, L. H., Adam, M. G., Kasthuriarachchi, N., and Lee, A. K. Y.: Characterization of carbonaceous aerosols in Singapore: insight from black carbon fragments and trace metal ions detected by a soot particle aerosol mass spectrometer, Atmos. Chem. Phys., 20, 5977-5993, 2020.

Wang, J., Ye, J., Liu, D., Wu, Y., Zhao, J., Xu, W., Xie, C., Shen, F., Zhang, J., Ohno, P. E., Qin, Y., Zhao, X., Martin, S. T., Lee, A. K. Y., Fu, P., Jacob, D. J., Zhang, Q., Sun, Y., Chen, M., and Ge, X.: Characterization of submicron organic particles in Beijing during summertime: comparison between SP-AMS and HR-AMS, Atmos. Chem. Phys., 20, 14091-14102, 2020.

Wang, J., Zhang, Q., Chen, M., Collier, S., Zhou, S., Ge, X., Xu, J., Shi, J., Xie, C., Hu, J., Ge, S., Sun, Y., and Coe, H.: First Chemical Characterization of Refractory Black Carbon Aerosols and Associated Coatings over the Tibetan Plateau (4730 m a.s.l), Environ. Sci. Technol., 51, 14072-14082, 2017.

Wang, J. F., Liu, D. T., Ge, X. L., Wu, Y. Z., Shen, F. Z., Chen, M. D., Zhao, J., Xie, C. H., Wang, Q. Q., Xu, W. Q., Zhang, J., Hu, J. L., Allan, J., Joshi, R., Fu, P. Q., Coe, H., and Sun, Y. L.: Characterization of black carbon-containing fine particles in Beijing during wintertime, Atmospheric Chemistry and Physics, 19, 447-458, 2019.

Wang, J. F., Wu, Y. Z., Ge, X. L., Shen, Y. F., Ge, S., and Chen, M. D.: Characteristics and sources of ambient refractory black carbon aerosols: Insights from soot particle aerosol mass spectrometer, Atmos. Environ., 185, 147-152, 2018.

Willis, M. D., Lee, A. K. Y., Onasch, T. B., Fortner, E. C., Williams, L. R., Lambe, A. T., Worsnop, D. R., and Abbatt, J. P. D.: Collection efficiency of the soot-particle aerosol mass spectrometer (SP-AMS) for internally mixed particulate black carbon, Atmos. Meas. Tech., 7, 4507-4516, 2014.

Xie, C., Xu, W., Wang, J., Liu, D., Ge, X., Zhang, Q., Wang, Q., Du, W., Zhao, J., Zhou, W., Li, J., Fu, P., Wang, Z., Worsnop, D., and Sun, Y.: Light absorption enhancement of black carbon in urban Beijing in summer, Atmos. Environ., 213, 499-504, 2019a.

Xie, C., Xu, W., Wang, J., Wang, Q., Liu, D., Tang, G., Chen, P., Du, W., Zhao, J., Zhang, Y., Zhou, W., Han, T., Bian, Q., Li, J., Fu, P., Wang, Z., Ge, X., Allan, J., Coe, H., and Sun, Y.: Vertical characterization of aerosol optical properties and brown carbon in winter in urban Beijing, China, Atmos. Chem. Phys., 19, 165-179, 2019b.

Xu, L., Suresh, S., Guo, H., Weber, R. J., and Ng, N. L.: Aerosol characterization over the southeastern United States using high-resolution aerosol mass spectrometry: spatial and seasonal variation of aerosol composition and sources with a focus on organic nitrates, Atmos. Chem. Phys., 15, 7307-7336, 2015.

Xu, W., Sun, Y., Wang, Q., Du, W., Zhao, J., Ge, X., Han, T., Zhang, Y., Zhou, W., Li, J., Fu, P., Wang, Z., and Worsnop, D. R.: Seasonal Characterization of Organic Nitrogen in Atmospheric Aerosols Using High Resolution Aerosol Mass Spectrometry in Beijing, China, ACS Earth and Space Chem., 1, 673-682, 2017.

---

## Author Comment (AC3) · 16 Feb 2021

**Responses to the reviewers:**

**"Elemental analysis of Oxygenated Organic Coating on Black Carbon Particles using a Soot-Particle Aerosol Mass Spectrometer" by Mutian Ma et al.**

**Reviewer #3**

The manuscript "Elemental analysis of Oxygenated Organic Coating on Black Carbon Particles using a Soot-Particle Aerosol Mass Spectrometer" by Mutian Ma et al. presents fundamental work on the use of electron-ionization mass spectra for estimates of elemental analysis (EA) organic compounds. The manuscript focusses on the differences between the laser vaporizer in the SP-AMS and the thermal vaporizer in the predecessor AMS. The main goal of the manuscript is to present a revised parameterization for EA estimation. The work is excellent, and the presentation is outstandingly clear. I have a few short comments on the statistical presentation of the results, which the authors should be able to address easily. I recommend publication after these minor comments.

Response: We thank for the constructive comments from the reviewer. Our responses to specific comments are shown in blue color below:

Specific comments:

1. First, the major result of this manuscript is the I-A, SP parameterization from laboratory work. The graph which truly illustrates this result is Figure S3. Figure 4 shows the "old" method. I recommend that the authors combine Figures 4 and S3 into one 6 panel figure.

   Response: As suggested by the reviewer, previous Figure S3 has been moved to the manuscript and combined with Figure 4.

2. Second, the abstract discusses the "relative error of O:C" for the compounds measured in the lab. The manuscript explains that this is the "average relative error". I am not entirely clear how the average was calculated, but I believe this is the Root Mean Squared Error (RMSE) or Mean Absolute Error (MAE). The authors should specify this precisely.

   The RMSE or MAE is a measure of the bias of the calibration. The authors should also report the precision of the calibration. An excellent example of this is found in Reggente, Dillner, and Takahama (Atmos Meas Tech 2016, https://doi.org/10.5194/amt-9-441-2016), but the authors may prefer some other formulation. My first comment also addresses the importance of precision, because the scatter in Figures S3 and 4 shows this precision.

   Response: We use the average value of percentage error in the abstract and main text to show not only the magnitude of the discrepancy and also the underestimation caused by applying I-A method for data obtained from the LV scheme. The RMSE and MAE were calculated and has been reported in Table S5 in the revised supplementary information. The main text has been modified to connect our discussion from revised Figure 4 to Table S5.

   Section 3.5, page 9, lines 13-15: "As shown in Figure 4 panel d-f, the H:C, O:C and $OS_C$ values calculated by the I-A$_{sp}$ method are better aligned with the 1:1 line compared to those determined by the I-A method (Figure 4) with smaller root mean squared error (RMSE) reported in Table S5."

Table S5. Root mean squared error (RMSE) and mean absolute error (MAE) of I-A and I-A$_{sp}$ method

|  | I-A H:C | I-A O:C | I-A$_{sp}$ H:C | I-A$_{sp}$ O:C |
|---|---|---|---|---|
| RMSE | 0.45 | 0.3 | 0.37 | 0.21 |
| MAE | 0.38 | 0.23 | 0.31 | 0.14 |